# Epigenetic Alterations in Canine Malignant Lymphoma: Future and Clinical Outcomes

**DOI:** 10.3390/ani13030468

**Published:** 2023-01-29

**Authors:** Esperanza Montaner-Angoiti, Pablo Jesús Marín-García, Lola Llobat

**Affiliations:** Departamento Producción y Sanidad Animal, Salud Pública y Ciencia y Tecnología de los Alimentos, Facultad de Veterinaria, Universidad Cardenal Herrera-CEU, CEU Universities, 46115 Valencia, Spain

**Keywords:** biomarkers, canine lymphoma, demethylation, epigenetic, therapeutics

## Abstract

**Simple Summary:**

Canine malignant lymphoma is one of the most common neoplasias in dogs. Even though breed influences the prevalence and prognosis, most cases present resistance to traditional chemotherapy. Canine breeds such as Labrador and Golden Retriever often have a worse prognosis for this reason. This high resistance to anticancer drugs makes it necessary to search for other therapies and other therapeutic targets. In this sense, the epigenetic regulators of neoplasia could be a promising object of study in the search for new research advances.

**Abstract:**

Canine malignant lymphoma is a common neoplasia in dogs, and some studies have used dogs as a research model for molecular mechanisms of lymphomas in humans. In two species, chemotherapy is the treatment of choice, but the resistance to conventional anticancer drugs is frequent. The knowledge of molecular mechanisms of development and progression of neoplasia has expanded in recent years, and the underlying epigenetic mechanisms are increasingly well known. These studies open up new ways of discovering therapeutic biomarkers. Histone deacetylases and demethylase inhibitors could be a future treatment for canine lymphoma, and the use of microRNAs as diagnosis and prognosis biomarkers is getting closer. This review summarises the epigenetic mechanisms underlying canine lymphoma and their possible application as treatment and biomarkers, both prognostic and diagnostic.

## 1. Introduction

Lymphomas are a common type of neoplasia in dogs, with prevalence estimated at around 100 cases per 100,000 dogs [1]. This prevalence depends on different variants, including canine breed. Labrador Retriever, Rottweiler and Boxer are breeds that present high prevalence of lymphomas [2,3,4]. Several clinical presentations and morphological subtypes have been studied, and the World Health Organization (WHO) classifies canine malignant lymphomas (CML) into different subtypes, regarding histopathology and immunohistochemistry, with two large groups according to origin: B-cell neoplasms and T-cell and putative natural killer cell neoplasms [5]. The most common clinical presentation includes multicentric, mediastinal, abdominal (gastrointestinal, hepatic, splenic and renal), cutaneous, ocular, central nervous system and pulmonary lymphoma [6]. Prognosis depends on type of lymphoma and clinical presentation, among other factors. In general, T-cell lymphomas present lower remission and survival time rates than B-cell lymphomas [5,7]. These forms are similar to human lymphomas, which have specific genetic abnormalities, including epigenetic changes, used in prognosis, diagnosis and treatment [8]. However, studies on this are very limited in CML. This review aims to summarise the molecular changes related to CML, epigenetics mechanisms and possible applications as biomarkers and treatments.

## 2. Canine Malignant Lymphomas (CML)

### 2.1. Aetiology of Lymphomas

CML is the most common haematopoietic neoplasm diagnosed in dogs and represents the most managed neoplasia in veterinary medical oncology [6]. Many studies of comparative oncology have investigated the epidemiology of CML, due to the similarities between CML and human non-Hodgkin’s lymphoma (NHL). CML shares similar characteristics of NHL patients, having similar clinical presentation, molecular and immunophenotypic composition, diagnosis, therapeutic protocols and treatment response [9,10]. These similarities make the dog one of the best models for the study of this human neoplasia. This disease consists of a heterogeneous group of lymphoid malignancies, so a multifactorial aetiology is considered, including age, sex, race/breed, autoimmune diseases, immunosuppression, and environmental factors.

In terms of age, CML presents most commonly in dogs from six to nine years old [3,11], whereas in humans, the average age of diagnosis for NHL is 67 years, with 57% of diagnostic cases found in those over 65 years of age [12]. Other factors related to NHL and CML are gender/sex. In fact, the influence of hormone status on the risk of NHL development has been extensively studied in human oncology, where the studies show a clear predominance of this disease in men [13,14] and poor prognosis, with 60% more likely to die of the disease [12]. In dogs, as in humans, several studies have reported a decreased risk in intact females [11,15]. Two mechanisms have been proposed to explain the reduced rate of NHL in females. The first is the antiproliferative effect of oestrogen on lymphoid cells through oestrogen receptor β signalling [16], and the second is related to interleukin 6 (IL6) serum levels. Low gene expression and IL6 protein production in peripheral blood mononuclear cells are related to low levels of 17β-oestradiol [17]. Considering that high serum levels of IL6 have been related to complete response and overall survival time [14,18], the elevated prevalence of NHL and poor prognosis in men could be explained by low levels of oestrogen.

In human oncology, race and ethnicity are considered risk factors for the development of NHL. Some studies have reported that white and non-Hispanic people are at higher risk of NHL than Asian/Pacific Islander, American Indian and black populations in the USA [12]. As in humans, canine breed is a relevant factor in CML. In fact, CML is over-represented in Labrador Retriever, Doberman, Rottweiler, Boxer, Bernese Mountain and Bullmastiff breeds [4]. For other breeds such as the Golden Retriever, the high prevalence has been reported only in the USA and Japan [19,20]. The genetic homogeneity of canine breeds facilitates the genetic studies related to lymphoma in dogs. A relation between canine breed and lymphoma immunophenotype has also been identified. Boxer, Shi-Tzu, Siberian Husky and Welsh Corgi have been linked to an increase of T-cell lymphoma, whereas Doberman, Rottweiler and German Shepherd have been related to B-cell lymphoma [4,21]. This breed predisposition suggests a genetic component in the pathogenesis of lymphoma. Specific mutations would explain the different prevalence of CML according to the canine breed. In one study, Modiano et al., (2005) identified patterns of chromosomal gains and losses as corresponding to the occurrence of B-cell and T-cell tumours in the Golden Retriever, a breed with a prevalence ratio of 1:1 B-cell/T-cell CML [3].

Other factors such as autoimmune disease and environmental characteristics have been correlated with NHL and CML. For example, several autoimmune diseases, such as Sjogren’s syndrome, systemic lupus erythematosus, celiac disease and scleroderma and immunosuppression due to infections or immunosuppressive therapy, have been associated with various subtypes of NHL [12]. In line with these results, some studies in veterinary medicine have reported high frequency of autoimmune diseases in dogs with CML [6]. Finally, the increased incidence of lymphoma in both humans and dogs suggests that environmental common risk factors might play a role in NHL and CML development. Some of the risks identified in the literature are exposure to chemicals and tobacco, living in industrial or farm areas and living near waste incinerators, radioactive or polluted sites [6,22]. Considering the environmental exposure of humans and their canine companions to be the same, the identification of canine genes that play a role in the pathogenesis of CML and its relationship with these risk factors may be useful for the identification of human cancer-associated genes and vice versa.

### 2.2. Diagnosis, Treatment and Prognosis of CML

The clinical signs associated with canine lymphoma are variable. In multicentric lymphoma, the most common form, there is usually generalised peripheral lymphadenopathy, painless, rubbery, hepatosplenomegaly and bone marrow involvement. Nonspecific signs such as anorexia, weight loss, vomiting, diarrhoea, emaciation, ascites, dyspnoea, polydipsia, polyuria and fever can appear [23]. However, the clinical features depend on the lymphoma presentation (Table 1).

In clinical practice, the most common method for diagnosis of CML is by cytological examination of fine-needle aspirate from a neoplastic lymph node. Fine-needle aspiration is a cheap, quick, sensitive, and minimally invasive technique, making it the diagnostic method of choice for high-grade canine lymphomas. When the cytological examination is non-conclusive or needs to be confirmed, the histopathology study of an incisional, excisional or true-cut biopsy should be performed [9]. Then, lymphoma is classified based on histological and cytological tests, according to the WHO and Kiel classification, respectively (Figure 1) [5,27,28].

In recent years, immunophenotyping has been carried out by immunohistochemistry, molecular methods to detect antigen receptor rearrangement (PARR) and flow cytometry to improve the first-choice treatment, according to this information. The latter method (flow cytometry), using fine-needle aspirates, is a minimally invasive method and assures rapid results. It is also an excellent tool for staging lymphoma, as it can identify neoplastic cell infiltration in different tissues, and one of the most recent and promising uses is the identification of minimal residual disease in peripheral blood, bone marrow and lymph nodes at the end of the chemotherapeutic protocol [30]. When flow cytometry or immunohistochemistry are not possible, the use of molecular techniques to detect antigen receptor rearrangement (PARR) can be used for diagnosing, staging and immunophenotyping in CML [31,32] in different samples, including fresh frozen tissue, formalin-fixed paraffin-embedded tissue, flow cytometry pellets and air-dried fine-needle aspirates [33]. This method is based on the fact that the lymphoma is a clonal lymphocyte expansion, so the neoplastic populations can be identified by detecting the presence of clonal populations of lymphocytes. This detection is carried out by amplification of the variable regions of immunoglobulin genes and T-cell receptor genes [34].

Multiagent protocols are the most used for the treatment of CML. The Wisconsin–Madison (CHOP) protocol, which combines cyclophosphamide, doxorubicin, vincristine and prednisolone, has the highest response rate and longest response durations [35]. Chemotherapy protocols follow two phases: the induction phase, which is a more intensive protocol that aims to induce complete remission, and in some cases, the clinician may choose to follow a less invasive maintenance protocol to maintain the remission status [6]. The recommendation to continue a maintenance chemotherapy protocol has been withdrawn, as it did not provide any treatment benefit after an induction protocol [35]. Another rescue protocol with good results is the LAP protocol, which combines lomustine, L-asparaginase and prednisone. The overall response with this rescue protocol has been estimated at around 77%, with 65% being a complete response [36].

In the event of a relapse, different options are available, and the treatment choice will depend on the moment of relapse in relation to the original protocol and the individual clinician’s preference. The common protocol of treatment for CML is resumed in Figure 2. Briefly, if the relapse occurs during the first-line induction protocol, it may indicate drug resistance to the protocol used. In this case, the rescue protocol will require alternative drugs not included in the first protocol. On the contrary, if the relapse occurs following completion of the first-line protocol, sometimes the clinical decision is to repeat the first-line protocol or change to a different rescue protocol. In this scenario, the use of drugs from the original protocol is possible [37,38].

Unfortunately, the rescue protocol typically results in lower response rates and shorter response durations and in some cases presents more toxicity than the first-line protocol. Given the high prevalence of this neoplasm in dogs, its poor prognosis, and the high toxicity of chemotherapy drugs, mainly those used as rescue treatment, progress in new treatments and biomarkers, both prognostic and diagnostic, is increasingly important. The study of the molecular mechanisms that underlie the development of CML and the response to treatment is essential to improve both the diagnosis and the prognosis of patients.

## 3. Epigenetic Mechanisms and CML

### 3.1. Histone Modifications

Histone modifications, such as acetylation, phosphorylation and methylation, could change chromatic structure and affect transcription and gene expression [39]. Briefly, these modifications could activate or inactivate heterochromatin and euchromatin, respectively, adding methyl, acetyl, and phosphate groups.

Histone methylation is carried out by specific methylases and demethylases on the residues of lysine and arginine [39,40]. Histone acetyltransferases (HAT) catalyse the transfer of an acetyl group of lysine, whereas histone deacetylases (HDAC) catalyse the reverse process [41]. Finally, histone kinases and phosphatase remove or add phosphate groups in the serine, threonine and tyrosine residues [42]. These changes modify the condensation of chromatin and facilitate or prevent gene expression (Figure 3).

Histone modifications are related to regulation of different biological processes, including development, genomic imprinting and HOX gene expression [44]. Regarding tumour development and progression, several histone modifications have been found. For example, functional dysregulation of histone acetyltransferases has been associated with cancer development, including B-cell [45,46] and T-cell lymphomas [47,48]. In humans, histone modifier gene mutations have been associated with inferior progression-free survival time in patients with T-cell lymphomas [48], and it has been demonstrated that HDAC regulates the expression of BCL6, involved in cell survival and/or differentiation in B-cell lymphoma [49], and HDAC inhibitors have been tried as therapy in humans [50,51,52]. In CML, repression of p16 tumour suppressor gene is regulated by histone H3 acetylation in vitro [53].

### 3.2. DNA Methylation

The addition of methyl groups to the cytosine CpG island of DNA is the most studied mechanism to regulate different processes, including chromosomal stability, genomic imprinting and X-chromosome inactivation [43]. DNA methylation is carried out primarily by three conserved enzymes in mammals: DNA methyltransferases (DNMT) 1, 3a and 3b [54], with different functions. DNMT1 maintains the methylation patterns during DNA replication and is expressed in somatic tissues and proliferating cells [55]. DNMT3a and 3b are de novo methyltransferases, and alternative transcripts with different functions are present in humans and mice [56]. As shown in Figure 4, DNMT adds methyl groups in the CpG island of promoter gene, preventing the RNA polymerase union and repressing the gene expression.

The relationship between DNA methylation aberrations and cancer has already been demonstrated in humans (see review [58]). Hypomethylation of several regions provokes the expression of oncogenes, whereas hypermethylation favours the development of cancer through the silencing of tumour suppressor genes. In dogs, the relationship between hypomethylation in genomic regions has been shown in canine leukaemia and lymphoma, where 30 and 69% of cases, respectively, were found. They are related to early phases of tumour transformation and progression [59]. In canine skin tumours, hypomethylation has been correlated to aggressiveness [60]. Hypomethylation has been found in canine osteosarcoma and lung cancer [61], whereas hypermethylation has been related to lymphoma [62], melanoma [63] and leukaemia [64], among other canine tumours, including canine mammary tumours. For example, hypomethylation of the intron region of the PAX6 motif of *CDH2* and *ADAM19* genes and hypermethylation at the PAX5 motifs in the intron regions of *CDH5* and *LRIG1* genes have been related to breast cancer [65].

Different GWAS (genome-wide association studies) have been carried out with relevant results. Specifically, in gastrointestinal lymphoma, 773 CpG islands are methylated, including promoter regions of 61 genes, including homeobox genes [66], whereas in high-grade B-cell lymphoma, Hsu et al., (2021) found 14 specific hypermethylated genes in samples of sick dogs but not in healthy dogs [67]. In canine diffuse large B-cell lymphoma, 1194 target loci are hypermethylated, including *HOX*, *BMP* and *WNT* genes, in promoter, 5′-UTRs upstream and exonic regions [68]. Hypomethylation in several regions and genes has been correlated with CML. Specifically, in dogs with NHL, higher DNA hypomethylation has been observed in circulating leukocytes [69].

In CML, not only methylated profiles have been studied. Different genes with relevant functions present methylation or non-methylation. In vitro studies have shown hypermethylation in promoter regions of *TWIST2* (related to transcriptional regulation [70]) and *TLX3* (orphan homeobox) genes in canine lymphoma cell lines [71]. Bryan et al., (2009) found abnormal hypermethylation of the CpG island in the *DLC1* (tumour suppressor gene) promoter in canine NHL samples, although it is not associated with silencing expression gene or survival [62]. Other tumour suppressor genes are inhibited by hypermethylation in CML. For example, tissue factor pathway inhibitor-2 (*TFPI-2*) is hypermethylated in its promoter region, decreasing its expression [72], and death-associated protein kinase *(DAPK)* hypermethylation has been correlated to negative prognosis in canine B-cell lymphoma [73].

One of the most widely studied tumour suppressor genes in haematological neoplasia and others in different species is the *p16* gene. The p16 protein inhibits the activity of cyclin-dependent kinase, as a negative control of the cell cycle to prevent phosphorylation of the retinoblastoma (pRb) protein. In fact, a high frequency of p16 mutations has been detected in many primary tumours and has been associated with progression of disease [74]. Related to CML, the inactivation of this gene by its CpG islands’ hypermethylation has been observed in canine lymphoid tumour cell lines [75], although results in vivo have demonstrated that p16 methylation status did not influence the prognosis [76]. Maylina et al., (2022) found a correlation between methylation of p16 gene, gene expression downregulation and hyperphosphorylation of pRb in canine lymphoma cell lines [77], and Fosmire et al., (2007) stated that inactivation of the *p16* gene by methylation occurs in high-grade T-cell NHL but not in B-cell [78]. Another tumour suppressor gene, DAPK, has been found hypermethylated in canine nodal high-grade B-cell lymphomas [79,80]. Recently, silencing of tumour suppressor genes *CIDEA*, *MAL* and *PCDH17* by hypermethylation in canine diffuse large B-cell lymphoma in vitro has been demonstrated [81].

Recent studies indicate that hypermethylome in canine B-cell lymphoma is conserved, unlike in humans [82], so these studies could improve the knowledge related to the molecular mechanism of CML and could open new lines of research to improve both treatments and prognostic and diagnostic biomarkers.

### 3.3. MicroRNAs (miRNAs)

MicroRNAs are small non-coding RNA molecules of 20–24 base pairs, which are unable to code for proteins, with a crucial role in the regulation of gene expression and cellular processes [83,84]. The scheme of microRNAs is shown in Figure 5. Briefly, miRNA is transcribed by RNA polymerase II as the primary transcript (pri-miRNA) in the nucleus, and the enzyme Drosha cuts the pri-miRNA to form a pre-miRNA (70–90 nucleotides). Later, Exportin 5 (XPO5) transports pre-miRNA from the nucleus to the cytoplasm, where it is cut by the Dicer enzyme (regulated by XPO5), to form a mature and short double-stranded miRNA. One of these chains is degraded, and a single strand of miRNA is incorporated into the RISC complex (RNA-induced silencing complex). This complex is related to the silencing of mRNA and regulation post-transcriptional gene expression via inhibition of protein translation or destabilization of target transcripts [85,86,87,88,89,90,91].

These small molecules of RNA are present both in intragenic and intergenic genome regions, are phylogenetically conserved [93] and have a crucial role in gene expression regulation and cellular processes, including neoplasia development [94,95]. In this sense, microRNAs are different target genes which have been related to proliferation, tumour initiation, metastasis, cell migration and invasion, senescence, cell cycle control, apoptosis and autophagy (Figure 6) [96,97,98,99,100,101,102].

Several miRNAs have been related to lymphoma in humans. MiR-150 controls expression of c-Myc transcription factor [103], whereas miR-155 promotes lymphoma progression by the PI3K/AKT pathway regulation [104]. Other miRNAs related to B-cell lymphoma are miR-17, miR-18, miR-19, miR-21, miR-92 and miR-217 as oncomiRNAs and miR-181, miR-144, miR-27, miR-34a, miR-28, miR-145 and miR-146 as tumour suppressor miRNAs [105]. In T-cell lymphoma, several miRNAs have been associated with disease progression, such as miR-187, miR-17, miR-135, miR-155, miR-16, miR-29, miR-96, miR-146, miR-101, miR-30 and miR-21 [106]. Several trials with miRNAs inhibitors are being used as a treatment for lymphomas. Currently, phases I of miR-34 mimic (MRX34) [107] and anti-miR-155 (MRG-106) [108] have been finished for lymphoma treatment.

In dogs, some miRNAs have been studied. Craig et al., (2019) carried out an interesting study on canine multicentric lymphoma, where they demonstrated the relationship between altered expression of 16 miRNAs in lymph nodes and 15 miRNAs in plasma for B-cell lymphoma and 9 miRNAs in lymph nodes and 3 miRNAs in plasma for B-cell and T-cell lymphoma, respectively [109]. Specifically, these authors found a relationship between the miR-181b, -181a, -31, -99a, -146a, -30b and 130b with B-cell lymphoma diagnosis and miR-182, -143, -130b, -183, -450a, -450b, -181b and -23a with T-cell lymphoma diagnosis. Differential expression of miRNAs has been found in canine intestinal large B-cell lymphoma, so tumour suppressor miRNAs such as miR-194, miR-192, miR-141 and miR-203 are downregulated, whereas oncogenic miRNAs such as miR-106a are upregulated [110].

## 4. New Treatments and Biomarkers Based on Epigenetics in CML

### 4.1. Histone Deacetylase (HDAC) Inhibitors as Treatment

HDAC inhibitors (HDACi) are classified based on their chemical structure in four categories, such as carboxylate, benzamide, hydroxamate and cyclic peptide. In humans, hydroxamate-acid-based vorinostat (SAHA) was approved by the FDA in 2006 [111], whereas cyclin tetra peptide-based romidepsin and belistat were approved in 2011 [112] and 2014 [113] for the T-cell lymphoma, respectively. In 2015, Panobinostat was approved by the FDA and EMA for multiple myeloma treatments [114,115], and it is currently being studied in a phase II trial for lymphoma treatment [116], and mocetinostat is awaiting EMA approval for Hodgkin’s lymphoma [117].

The use and effectivity of HDACi in CML have been demonstrated in in vitro experiments. Dias et al., (2018) studied seven different HDACi in canine lymphoma cell lines, and their results indicated that Panobinostat is the most effective and with fewer toxicity compounds of the nine evaluated [118]. Two hydroxamate-based HDACi (Suberoylanilide hydroxamic acid-SHANA and Panobinostat) have shown demonstrated efficacy and low toxicity in dogs, and Zhang et al., (2016) evaluated the toxicity of HZ1006 (other hydroxamate-base HDACi) in dogs, with promising results [119]. However, the number of in vivo studies on the effectivity and toxicity of HDACi in dogs is very limited, so further studies of different HDACi as CML therapies are necessary to confirm the use of these molecules in this canine neoplasia.

### 4.2. Demethylation and Deacetylation Drugs as Treatment for Canine Lymphoma

Different demethylated agents have been tested for lymphoma treatment in humans. Inhibitors of lysine-specific demethylase 1 (LSD1) are some of the most studied. LSD1 regulates the transcription genes, repressing transcription by demethylation of histone H3 lysine 4 (H3K4) [120] and activating transcription by demethylation of histone 3 lysine 9 (H3K9) [121]. Different clinical trials are being carried out to evaluate the efficacy and toxicity of various LSD1 inhibitory molecules. Domatinostat (4SC-202) is an HDACi and LSD1 inhibitor, which induced cell death in T-cell lymphoma cell lines in vitro [122]. Hollebecque et al., (2022) showed the results of a phase I clinical trials with an oral LSD1 inhibitor (CC-90011) in NHL, indicating that this molecule presents a favourable tolerability profile, good clinical activity and durable responses [123]. The same research group has demonstrated that LSD1 expression in NHL is associated with chemoresistance [124], so the use of LSD1 inhibitors could be a new approach in drug-resistant lymphomas. In fact, LSD1 inhibitor GSK2879552 showed results in the phase I trial to relapsed/refractory myelodysplastic syndromes treatment [125]. Other LSD1 inhibitors are being tested. For example, ZY0511 is a potent LSD1 inhibitor, which induces the methylation of H3K4 and H3K9 in B-cell lymphoma, promoting apoptosis in both in vitro and in vivo xenograft experiments [126]. Regarding chemoresistance, high levels of inhibitor of histone 3 Lys 27 (KDM6B) are associated with poor prognosis and survival in patients with diffuse large B-cell lymphoma, and its inhibitor GSK-J4 induces apoptosis in cell lines with high levels of KDM6B [127]. These results suggest that these inhibitors could be the treatment of choice in lymphomas resistant to other conventional treatments. Other molecules related to lymphoma progression have been found, such as histone methylation inhibitors, including PKF118-310 (a KDM4A inhibitor) [128] and histone H3 Lys 27 demethylase [129], which could be used in future lymphoma treatments.

In CML, some methylation regulators have also been tested and could be a new possible treatment. For example, antimetabolite 6-thioguanine (6-TG) causes downregulation of DNA methyltransferase 1 in canine lymphoma cell lines, increasing cell survival [130]. Recently, Itoh et al., (2021) demonstrated that olsalazine reduces DNA methylation, including the methylation in the promoter regions of ADAM23, FES and CREB3L1 genes, inhibiting cell proliferation in canine lymphoid tumour cell lines [131]. Similar results have been found with azacytidine and decitabine, two hypomethylating drugs. However, in vivo studies in mice indicated that these drugs did not arrest tumour growth, although the methylation in the target gene promoter was reduced [132]. Sloan et al., (2021) showed an overexpression of protein arginine methyltransferase 5 (PRMT5) in primary B-cell canine lymphomas, arguing that PRMT5 inhibitors could be a candidate as a new therapy in CML [133].

Likewise, methylation has been related to chemoresistance in CML, so demethylated drugs are interesting as candidate therapy in refractory and/or chemo-resistant CML. Several studies have observed differences between drug-resistant cell methylation patterns. Thus, the CpG island in the upstream region of the exon 2 in the ABCB1 gene is hypermethylated (and presents low expression) in drug-sensitive canine lymphoma cell lines, whereas it is hypomethylated in drug-resistant canine lymphoma cell lines, and these results have been corroborated in in vivo studies [134]. Furthermore, O-methyl guanine DNA methyltransferase expression has been correlated to lomustine resistance in canine lymphoma cell lines [135], and canine cell lines resistant to doxorubicin show hypermethylation in gene promoter regions and low expression of 24 differentiated genes [67]. These results indicate that, as in humans, demethylating drugs could be interesting to use in CML with poor prognosis or resistant to chemotherapy.

Regarding deacetylation drugs, a multicentric study in canine lymphoma shows that treatment with sulforaphane, an isothiocyanate derived from some vegetables, is associated with change in proteome [136], although its results in clinical practice are as yet unknown. However, the effect of sulforaphane as an HDAC and DNMT inhibitor has been proven in human cancer prostate cells [137], acute lymphoblastic leukaemia cells [138] and multiple myeloma [139]. Other HDAC inhibitors have been studied as potential canine lymphoma treatments, such as class I HDAC inhibitors [140] or BET and SYK inhibitors [141,142].

### 4.3. microRNAs as Biomarkers and Treatment in CML

Alteration in miRNA expression has been related to different human diseases [143], including malignant tumours, where miRNAs have relevant roles in proliferation, apoptosis, invasion, metastasis and angiogenesis [144]. Several miRNAs have been related to CML and some of them have been postulated as possible biomarkers and/or therapeutic targets [145]. Two of the most studied miRNAs in CML are miR-155 and miR-17-5p. The former, miR-155, is downregulated in canine diffuse large B-cell lymphoma [146], whereas miR-17-5p is upregulated in lymph nodes of dogs with canine B-cell lymphomas [147]. Moreover, the molar ratio between these miRNAs in plasma is correlated with the WHO gradient classification [148], so this ratio could be used as a minimally invasive biomarker for diagnostic classification. miR-181 is an miRNA with tumour suppressor functions, and it is also downregulated in the lymph nodes of dogs with T-cell lymphomas [149]. Other miRNAs with tumour suppressor functions such as miR-203 and miR-218 present low expression in canine B- and T-cell lymphomas, as well as in humans, whereas miRNAs with oncogene upregulation function, such as miR-19a, miR-19b and miR17-5p, present high expression [147]. These results show that inhibitors of these miRNAs could be used in new treatments or as biomarkers in CML. Elshafie et al., (2021) observed an upregulation and downregulation of miR-34a and let-7 family miRNAs, respectively, with their results suggesting that the latter family of miRNAs alone can discriminate neoplastic canine diffuse B-cell lymphoma and non-neoplastic tissue in 97% of cases [146]. However, not only these miRNAs have been postulated as possible biomarkers in CML. The serum expression levels of let-7b, miR-223, miR-25 and miR-92a and mi-423a present reduced expression levels in dogs with lymphoma compared to healthy dogs. Specifically, miR-25 levels are related to histological grade in CML [150]. To determine the possible use of some miRNAs as biomarkers, Craig et al., (2019) analyzed the expression of 38 miRNAs both in plasma and lymph nodes in multicentric B- and T-cell lymphoma compared to healthy dogs. Their results showed altered expression in lymph nodes for 16 and 9 miRNAs in B- and T-cell lymphoma, respectively. Different miRNAs presented altered expression in plasma samples, without correlation with lymph node results. In fact, results in plasma show 15 and 3 miRNAs with altered expression in B- and T-cell lymphoma, respectively. In this study, the authors correlated expression levels of miRNAs with progression of diseases, correlating low expression of 10 miRNAs with progression-free survival and 3 with overall survival [109]. Table 2 summarises the miRNAs related to CML found in this study and others, the target gene of miRNAs, possible function and therapeutic applications.

Some altered expression miRNAs have been related to a specific clinical presentation of CML. High serum levels of miR-122 have been found in CML involving the liver [166], whereas canine intestinal T-cell lymphoma presents specific expression patterns, including downregulation of miR-194, miR-192, miR-141 and miR-203 and upregulation of the miR-106a cluster [110].

In recent years, a new approach has been studied. Plasma levels of circulating extracellular vesicles (EV) and the miRNAs present in these EV are being postulated as new possible biomarkers. Indeed, the plasma EV population is altered in human and canine diffuse large B-cell lymphomas [167] and related to progression of disease [168]. In addition, exosomes differ in their miRNAs content in cell lines resistant to chemotherapy; thus, the content of miR-151, miR-8908a-3p and miR-486 in plasma exosomes is different between vincristine-sensitive and resistant canine lymphoid tumour cell lines [169]. These results promote new lines of research in the search for biomarkers in malignant lymphoma in both humans and dogs.

## 5. Conclusions

The traditional treatment consists of radio- and chemotherapy, with poor results in many cases. The epigenetic mechanisms underlying canine lymphoma are increasingly well known, which implies that their possible application both as new treatments and/or biomarkers for prognosis and diagnosis is getting closer. In humans, therapies based on histone deacetylase and demethylase inhibitors have been approved in recent years. MiRNAs have been used as biomarkers in other tumours, so their use and application in the near future will probably improve CML prognosis and early diagnosis.

## Figures and Tables

**Figure 1 animals-13-00468-f001:**
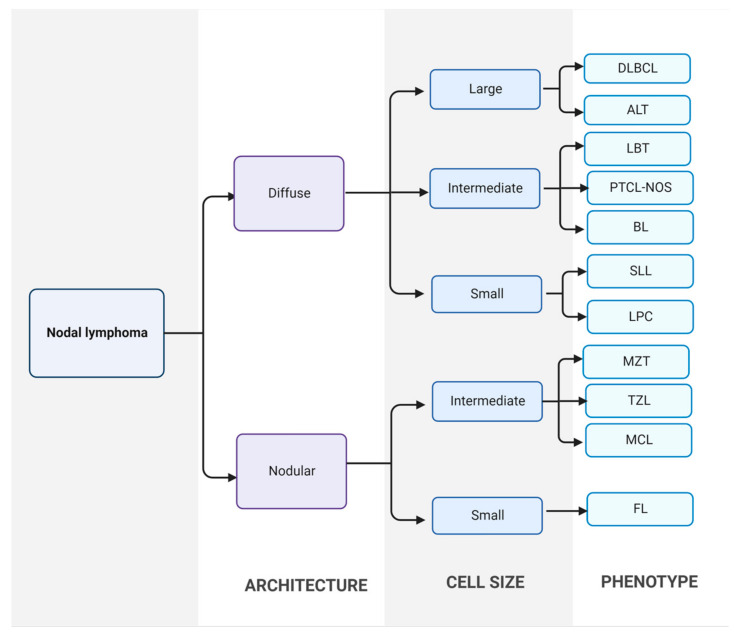
Classification of canine nodal lymphoma. First, lymphomas are divided into diffuse or nodular forms using excisional lymph node sections. Then, neoplastic populations are divided into large, intermediate, and small cells. Finally, final diagnosis is established. DLBCL: diffuse large B-cell lymphoma; ALT: anaplastic large T-cell lymphoma; LBT: T-cell lymphoblastic lymphoma; PTCL-NOS: peripheral T-cell lymphoma; BL: B-cell lymphoma; SLL: small lymphocytic lymphoma; LPC: lymphoplasmacytic lymphoma; MZL: marginal zone lymphoma; TZL: T-zone lymphoma; MCL: mantle cell lymphoma; FL: follicular lymphoma [29].

**Figure 2 animals-13-00468-f002:**
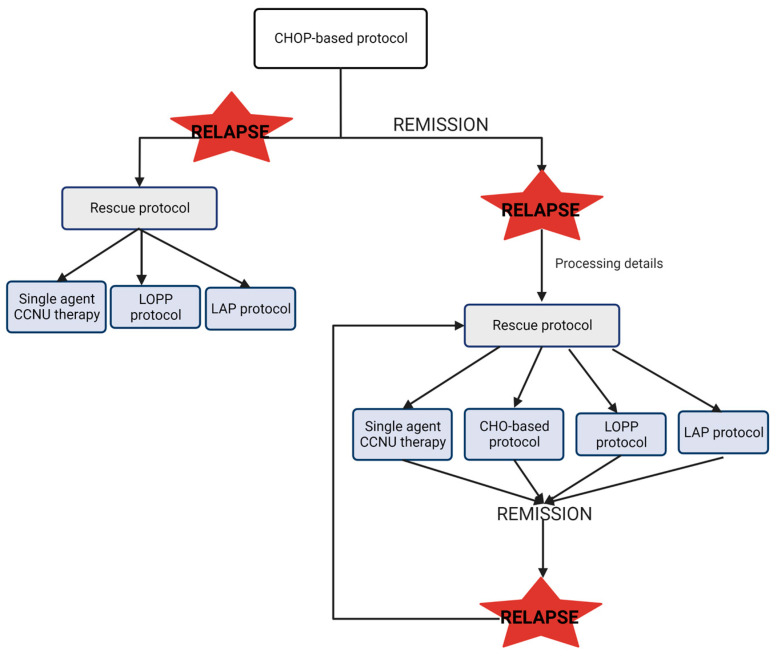
Scheme of common treatment protocol in CML. CHOP: Wisconsin–Madison protocol; CCNU: lomustine; LOPP: vincristine procarbazine and prednisolone protocol. LAP: lomustine, L-asparaginase and prednisone protocol [35,36].

**Figure 3 animals-13-00468-f003:**
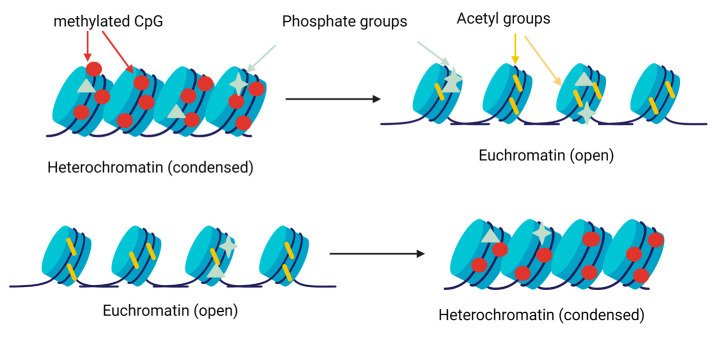
Scheme of the gene expression regulation by histone modifications. Methyltransferases add methyl groups in lysine and arginine residues, favouring chromatin condensation and preventing gene expression. Acetyltransferases and deacetylases add or remove acetyl group in lysine residues, decreasing or increasing, respectively, the condensation of chromatin. The kinases and phosphatases add phosphate groups in different locations, changing the condensation of chromatin [43].

**Figure 4 animals-13-00468-f004:**
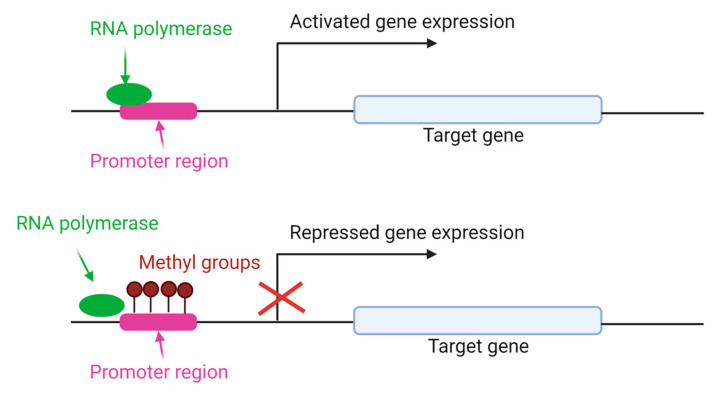
Gene expression regulated by DNA methylation. DNMT enzymes add methyl groups in cytosine residues of CpG island. As a result, RNA polymerase is unable to bind to the DNA sequence of the gene promoter and the gene expression is repressed [57].

**Figure 5 animals-13-00468-f005:**
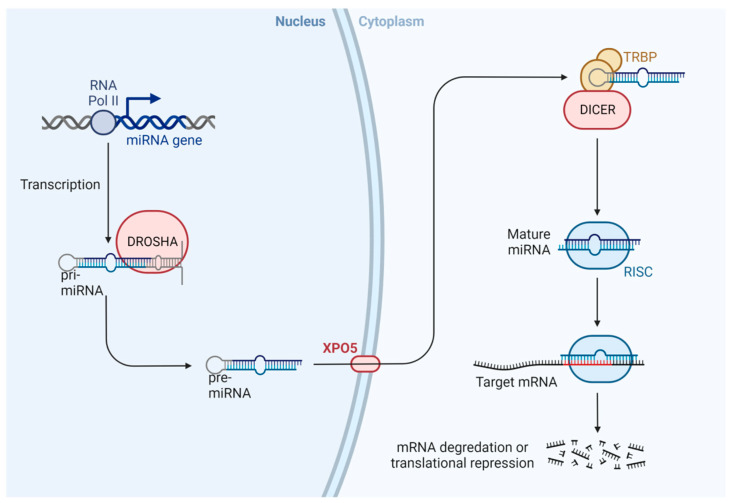
Scheme of miRNA biogenesis. In the nucleus, miRNA is transcribed by RNA polymerase II as primary transcripts (pri-miRNA). The Drosha enzyme cuts this pri-miRNA to form a pre-miRNA, which is actively transported to the cytoplasm by the nuclear transport receptor exportin 5 (XPO5). In the cytoplasm, the pre-miRNA is cut by a second enzyme, Dicer, to form a mature and short double-stranded miRNA molecule. The miRNA duplex is incorporated into the RISC protein complex [92].

**Figure 6 animals-13-00468-f006:**
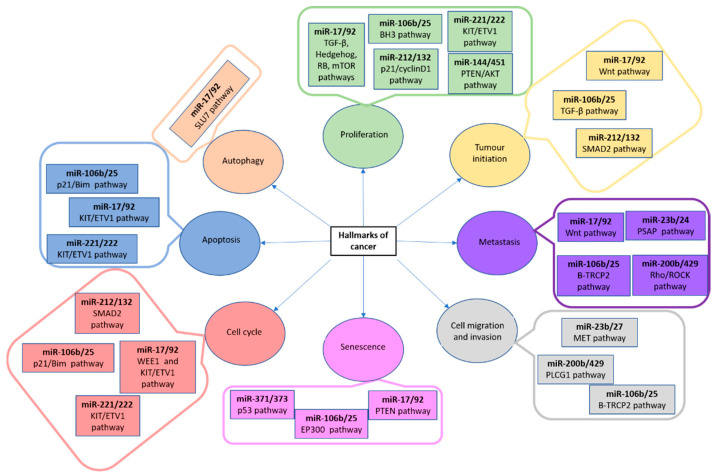
MiRNA clusters, their pathways and associated with cancer hallmarks. AKT: AKT serine/threonine kinase; BH3: Bcl-2 homology 3 domain; BIM: Bcl-2-like protein 11; EP300: E1A-asosciated protein p300; ETV1: Ets variant gene 1; KIT: proto-oncogene tyrosine-protein kinase; MET: MET proto-oncogene receptor tyrosine kinase; mTOR: mammalian target of rapamycin complex 1; p21: cyclin-dependent kinase inhibitor 1A; PLCG1: phospholipase c gamma 1; PSAP: prosaposin; p53: tumour protein p53; PTEN: phosphatase and tensin homolog; RB1: RB transcriptional corepressor 1; RHO: ROCK Rho-associated protein kinase; SLU7: pre-mRNA splicing factor SLU7; SMAD2: mothers against dpp homolog 2; TRCP2: F-box and WD repeat domain containing 11; TGF: transforming growth factor; WEE1: Wee1A kinase; Wnt: wingless-type mmtv integration site family [57].

**Table 1 animals-13-00468-t001:** Common clinical manifestations according to the type and presentation of the lymphoma.

Lymphoma Type	Clinical Signs	Observation	References
Multicentric	Generalised lymphadenomegaly. Involvement of the spleen, liver, and bone marrow.	The most predominant type of lymphoma (80–85%).	[6]
Gastrointestinal	Vomiting, diarrhoea, melena, and weight loss due to malabsorption and maldigestion of nutrients. It can present as an acute or chronic lymphoma.	The second most prevalent (10%).	[24]
Mediastinal	Coughing, breathing difficulty and pleural effusions. Polyuria and polydipsia can be seen due to hypercalcaemia (present in 40% of cases).	The third most common type of canine lymphoma.	[25]
Extranodal	Blindness, renal failure, seizures, bone fractures and respiratory disease. Organs typically affected are the eyes, kidneys, lungs, skin, and central nervous system; other areas that can be invaded include the mammary tissue, liver, bones and mouth. Cutaneous lymphoma, present as a skin rash with dry, red, and itchy bumps or solitary or generalised scaly lesions and can also affect the oral cavity, causing ulcers, lesions and nodules on the gums, lips and roof of the mouth.	The rarest form. Cutaneous lymphoma, the most common form of extranodal lymphoma, in early stages usually present as a skin rash with dry, red and itchy bumps or solitary or generalised scaly lesions. Sometimes may be mistaken for allergies or fungal infections. As the disease progresses, cutaneous lesions will become more severe, and large masses or tumours can develop. Cutaneous lymphoma can also affect the oral cavity causing ulcers, lesions and nodules on the gums, lips, and roof of the mouth.	[26]

**Table 2 animals-13-00468-t002:** MicroRNAs (miRNAs) related to canine malignant lymphoma (CML), target of these miRNAs, result of aberrant expression and possible uses.

miRNAs	Expression Profiles	Target Genes *	Result of Aberrant miRNA Expression	Possible Uses	References
miR-155	Downregulated	HDAC4, PIK3R1, SMAD5, SHIP1	Induces cell proliferation, activates oncogenic AKT signalling, enhances tumour aggressiveness	Potential therapeutic target	[148]
miR-17-5p	Upregulated	PTEN	Disease progression	Potential prognostic biomarker Potential therapeutic target	[148]
miR-181	Downregulated	Bcl-2/TCL-1	Activation of oncogenes	Potential prognosis biomarker Potential therapeutic target	[110]
miR-203	Upregulated	ABL1	Tumour cell proliferation	Potential therapeutic target	[110,147]
miR-218	Upregulated	SLIT2/3	Tumour cell proliferation	Potential therapeutic target	[151]
miR-19a	Upregulated	PTEN	Tumour cell proliferation	Potential prognostic biomarker Potential therapeutic target	[152]
miR-19b	Upregulated	PTEN	Tumour cell proliferation	Potential prognostic biomarker Potential therapeutic target	[152]
miR-34a	Upregulated	MET, SIRT1, CDK6, VEGF1	Induced cell migration	Potential prognostic biomarker Potential therapeutic target	[153]
Let-7 family	Downregulated	PRDM1 ⁄Blimp-1	Reduces terminal B-cell differentiation	Potential diagnosis biomarker Potential therapeutic target	[154]
miR-223	Downregulated	LMO2	Increase tumour progression	Potential prognosis biomarker	[155]
miR-25	Downregulated	LATS2	Tumour cell proliferation	Potential prognosis biomarker Potential therapeutic target	[156]
miR-92a	Downregulated	TET2	Tumour cell proliferation Increase tumour progression	Potential diagnosis biomarker Potential prognosis biomarker	[157]
miR-423a	Downregulated	Unknown	Tumour cell proliferation	Potential diagnosis biomarker Potential prognosis biomarker	[150]
miR-29a/b/c	Upregulated	MYC, DNMT3B, MCL1, BIM, CDK6, AKT, TCL1	Tumour B-cell proliferation	Potential diagnosis biomarker Potential therapeutic target	[158]
miR-31	Upregulated	E2F2, PI3KC2A	Tumour B-cell proliferation	Potential diagnosis biomarker	[159]
miR-23a	Downregulated	MYC, MTSS1	Tumour B-cell proliferation Tumour T-cell proliferation	Potential diagnosis biomarker	[160]
miR-26b	Downregulated	EZH2, COPS2, KPNA2, MRPL15, NOL12	Tumour B-cell proliferation Tumour T-cell proliferation	Potential diagnosis biomarker	[161]
miR-99a	Downregulated	Unknown	Tumour B-cell proliferation	Potential diagnosis biomarker	[162]
miR-125a	Downregulated	TNFAIP3, NF1, Bcl-2	Tumour B-cell proliferation	Potential diagnosis biomarker	[163]
miR-143	Downregulated	B7H1, Bcl-2	Tumour B-cell proliferation	Potential diagnosis biomarker	[164]
miR-145	Downregulated	ROCK1, MYC	Tumour T-cell proliferation	Potential diagnosis biomarker	[165]

* HDAC4: histone deacetylase 4; PIK3R1: phosphoinositiden-3-kinase regulatory subunit 1; SMAD5: SMAD family member 5; SHIP1 (INPP5D): inositol polyphosphate-5-phosphatase D; PTEN: phosphatase and tensin homolog; Bcl-2/TCL-1: Bcl-2/T-cell leukaemia/lymphoma protein 1; ABL1: ABL proto-oncogene 1, non-receptor tyrosine kinase; SLIT2/3: slit guidance ligand 2/3; MET: MET proto-oncogene, receptor tyrosine kinase; SIRT1: sirtuin 1; CDK6: cyclin-dependent kinase 6; VEGF1: vascular endothelial growth factor A; PRDM1 (Blimp-1): PR/SET domain 1 (B-lymphocyte-induced maturation protein 1); LMO2: LIM domain only 2; LATS2: large tumour suppressor kinase 2; TET2: tet methyl cytosine dioxygenase 2; MYC: MYC proto-oncogene, BHLH transcription factor; DNMT3B: DNA methyltransferase 3 beta; MCL1: MCL1 apoptosis regulator, Bcl-2 family member; BIM: Bcl-2-like protein 11; AKT: AKT serine/threonine kinase; TCL1: TCL1 family AKT coactivator A; E2F2: E2F transcription factor 2; PI3KC2A: Phosphatidylinositol-4-Phosphate 3-Kinase Catalytic Subunit Type 2 Alpha; MTSS1: metastasis suppressor 1; EZH2: enhancer of zeste 2 polycomb repressive complex 2 subunit; COPS2: COP9 signalosome subunit 2; KPNA2: karyopherin subunit alpha 2; MRPL15: mitochondrial ribosomal protein L15; NOL12: Nuclear protein 12; TNFAIP3: transforming necrotic factor alpha induced protein 3; NF1: neurofibromin 1; Bcl-2: B-cell lymphoma apoptosis regulator; B7H1 (PD-L1): programmed death-ligand 1, ROCK1: Rho-associated coiled-coil containing protein kinase 1.

## Data Availability

Not applicable.

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
