# Peer review of "Epigenetic Alterations in Canine Malignant Lymphoma: Future and Clinical Outcomes"

_animals, 2023, doi:10.3390/ani13030468_

Round 1

Reviewer 1 Report

[General concept Comment]

In this paper, the authors present their work as a review of epigenetics alterations in canine malignant lymphoma: future 2 and clinical outcomes.

The manuscript is relevant to the field of canine lymphoma, however, a bit incomprehensible and less structured, it could be better written for the future. The English language is less understandable, and it seems not to use scientific English. The reviewer strongly recommends the manuscript to be checked by a professional English editing service. The authors try to explain the information from the appropriate literature and present easy-to-read and understandable thanks to the tables and figures the authors included. Bibliographic references are adequate, and they include six articles published in 1994, 1997, 1998, and 1999. The remaining articles were published between 2000 and 2022. The conclusion could have been written better. This manuscript is an interesting review, however, needed major revision for the final version.

Following are specific comments regarding the manuscript:

[Specific Comment]

Line 14 [Abstract section]:

This sentence “Canine malignant lymphoma is a common neoplasia in dogs and humans,…..” seem to mention dogs only, not humans.

Line 28-30 [Introduction section]:

This sentence is very confusing, please remove '…being…' [Line 29] and turn it into a sentence that is simpler and easier to understand?

Line 32 [Introduction section]:

The authors mentioned CML as an abbreviation for canine malignant lymphomas. However, CML is generally used as chronic myelogenous leukemia. Please correct this point. The abbreviation referred to is also related to its use in Line 44 and many sentences in this manuscript.

Line 59:

The term NHL and CML are included into ‘lymphoma’ term in dogs. In other words, there is “canine NHL”. However, it will be different meaning if the authors mentioned ‘human NHL’. Please check the entire manuscript about the term “NHL”.

Line 66:

Do “Low expression and IL6 production” mean “Low gene expression and protein production of IL6”?

Line 86:

“…B-cee/T-cell….” is it mean “…B-cell/T-cell….”? What is “CMT”?

Line 87 and 88

“autoimmune disease” should be “autoimmune diseases”.

Line 89:

The use term “system lupus erythematosus” should be “systemic lupus erythematosus”

Line 109:

The use of left alignment is better in Table 1. The words in the table are very long and less understandable. Please add another column citing references.

Line 112:

The words “There is…..” seem not appropriate for the sentence.

Line 128:

Is “LBL” correct?

Line 148-155 and 168-175:

Please cite the references.

Line 166 and 167

In the figure, there are “LOOP”, but in the figure legend “LOPP”.

As the rescue protocol, LAP protocol (L- asparaginase, CCNU and prednisolone) was reported and often used in clinics. used. The reviewer recommends indicate the LAP protocol as rescue protocol.

Line 226-232:

This sentence is very long, therefore, it becomes confusing, please separate the sentences and turn them into a sentence that is simpler and easier to understand.

Line 233:

The authors mentioned “….GWAS…”, however, the meaning of the abbreviation could not be found.

Line 240 and 247:

Please consistently use NHL for the manuscript, not “….non-Hodgkin lymphoma….” or “….non-Hodgkin’s lymphoma….”.

Line 306-315:

There are explanations for miRNAs in humans, however, there are no appropriate references in dogs. Please add the explanation for the dogs.

Line 317-333:

The authors mentioned HAC and HACi as an abbreviation for histone deacetylases and histone deacetylases inhibitors. Is this abbreviation commonly used? The common use is HDAC and HDACi.

The abbreviation referred to is also related to its use in many sentences in Lines 326-333. Change them throughout the manuscript.

Line 329:

What is “….SHANA…”?

Line 336:

The title “Demethylation drugs and methylation profiles related to treatment” seem to be inappropriate, since the methylation processes occurs before using demethylation drugs.

Line 347:

“…..lymphoma non-Hodgkin…”

Please see the comment in Lines 240 and 247.

Line 421:

The use of left alignment is better in Table 2. Please add another column citing references.

Line 424

“Blc-2” should be “Bcl-2”.

Line 430

“Blc2” should be “Bcl-2”.

Line 455-457 [Conclusion section]:

Please remove the first sentence in the conclusion “Canine malignant lymphoma (CML) is a heterogeneous neoplasia which affects 455 around 100 dogs per 100,000, depending on the several factors, as canine breed.”

Author Response

Thank you very much for your comments and suggestions Following your recommendation, the manuscript has been revised by a native speaker (attach certificate). We have answered the rest of the questions one by one below.

Following are specific comments regarding the manuscript:

[Specific Comment]

Line 14 [Abstract section]:

This sentence “Canine malignant lymphoma is a common neoplasia in dogs and humans,…..” seem to mention dogs only, not humans.

Done.

Line 28-30 [Introduction section]:

This sentence is very confusing, please remove '…being…' [Line 29] and turn it into a sentence that is simpler and easier to understand?

Done.

Line 32 [Introduction section]:

The authors mentioned CML as an abbreviation for canine malignant lymphomas. However, CML is generally used as chronic myelogenous leukemia. Please correct this point. The abbreviation referred to is also related to its use in Line 44 and many sentences in this manuscript.

We disagree with the reviewer. According to Pubmed (http://allie.dbcls.jp/pair/CML;Canine+malignant+lymphoma.html) the correct abbreviation for Canine Malignant Lymphoma is CML. In fact, different papers use this abbreviation, as

https://pubmed.ncbi.nlm.nih.gov/9173362/

https://pubmed.ncbi.nlm.nih.gov/10349709/

https://pubmed.ncbi.nlm.nih.gov/22222006/

Line 59:

The term NHL and CML are included into ‘lymphoma’ term in dogs. In other words, there is “canine NHL”. However, it will be different meaning if the authors mentioned ‘human NHL’. Please check the entire manuscript about the term “NHL”.

The abbreviation NHL is used in the manuscript for non-Hodgkin lymphoma in humans, not for canine Non-Hodgkin lymphoma. the comparison carried out in the manuscript in this section refers to the similarities between CML (canine) and NHL (human), since both present numerous similarities in clinical aspects, among others.

Line 66:

Do “Low expression and IL6 production” mean “Low gene expression and protein production of IL6”?

Yes. To improve the understanding, we have change the sentence to “Low gene expression and protein production of IL6”.

Line 86:

“…B-cee/T-cell….” is it mean “…B-cell/T-cell….”? What is “CMT”?

Sorry for these mistakes. We have change to B-cell and CML.

Line 87 and 88

“autoimmune disease” should be “autoimmune diseases”.

Done.

Line 89:

The use term “system lupus erythematosus” should be “systemic lupus erythematosus”

Done.

Line 109:

The use of left alignment is better in Table 1. The words in the table are very long and less understandable. Please add another column citing references.

Done.

Line 112:

The words “There is…..” seem not appropriate for the sentence.

The sentence has been changed to “Fine-needle aspirate is a cheap,…”.

Line 128:

Is “LBL” correct?

Sorry for this mistake. LBL has been change to “LBT”.

Line 148-155 and 168-175:

Please cite the references.

Done.

Line 166 and 167

In the figure, there are “LOOP”, but in the figure legend “LOPP”.

As the rescue protocol, LAP protocol (L- asparaginase, CCNU and prednisolone) was reported and often used in clinics. used. The reviewer recommends indicate the LAP protocol as rescue protocol.

Thank you for your recommendation. LOOP has been changed to LOPP in the figure, and the LAP protocol has been added in the figure 2 and in the text (lines 155-158).

Line 226-232:

This sentence is very long, therefore, it becomes confusing, please separate the sentences and turn them into a sentence that is simpler and easier to understand.

Following your recommendation, these sentences have been shortened for ease of understanding.

Line 233:

The authors mentioned “….GWAS…”, however, the meaning of the abbreviation could not be found.

The meaning of the abbreviation GWAS (genome-wide association studies) has been added in the manuscript.

Line 240 and 247:

Please consistently use NHL for the manuscript, not “….non-Hodgkin lymphoma….” or “….non-Hodgkin’s lymphoma….”.

Non-Hodgkin’s lymphoma has been changed to non-Hodgkin lymphoma in all the manuscript.

Line 306-315:

There are explanations for miRNAs in humans, however, there are no appropriate references in dogs. Please add the explanation for the dogs.

The information about miRNAs in dogs has been added (lines 321-330).

Line 317-333:

The authors mentioned HAC and HACi as an abbreviation for histone deacetylases and histone deacetylases inhibitors. Is this abbreviation commonly used? The common use is HDAC and HDACi.

The abbreviation referred to is also related to its use in many sentences in Lines 326-333. Change them throughout the manuscript.

Sorry for this mistake. HAC and HACi have been changed to HDAC and HDACi respectively in all the manuscript.

Line 329:

What is “….SHANA…”?

SHANA is Suberoylanilide hydroxamic acid. This information has been added.

Line 336:

The title “Demethylation drugs and methylation profiles related to treatment” seem to be inappropriate, since the methylation processes occurs before using demethylation drugs.

Thank you very much for your recommendation. The title has been changed to “Demethylation and deacethylation drugs as treatment to canine lymphoma”.

Line 347:

“…..lymphoma non-Hodgkin…”. Please see the comment in Lines 240 and 247.

Done.

Line 421:

The use of left alignment is better in Table 2. Please add another column citing references.

Done.

Line 424

“Blc-2” should be “Bcl-2”.

Done.

Line 430

“Blc2” should be “Bcl-2”.

Done.

Line 455-457 [Conclusion section]:

Please remove the first sentence in the conclusion “Canine malignant lymphoma (CML) is a heterogeneous neoplasia which affects 455 around 100 dogs per 100,000, depending on the several factors, as canine breed.”

Done.

Reviewer 2 Report

Thank you very much for submitting the manuscript being entitled "Epigenetics alterations in canine malignant lymphoma: future and clinical outcomes" by Esperanza Montaner-Angoiti, Pablo Jesús Marín-García, and Lola Llobat. As stated in the abstract, the presented review summarizes the epigenetic mechanisms underlying canine lymphoma and its possible application as treatment and biomarkers, both prognostic and diagnostic.

Although I acknowledge that the submitted manuscript will be of interest to readers of Animals, their manuscript is not acceptable in its current form, but may be considered for publication after major revision.

As a result, the authors will need to revise as follows:

(1) Prior to thoroughly reviewing the present draft, I have to ask the authors to cross-check the below listed references for potential supplementation to the manuscript.

Epigenetics in canine lymphoma

PMID: 35955829, PMID: 35876604, PMID: 35409379, PMID: 34954594, PMID: 34570764, PMID: 32926463, PMID: 32154977, PMID: 28111882, PMID: 27747218, PMID: 27630997, PMID: 29061940, PMID: 25716962, and PMID: 25108839.

Oncogenes, tumor markers and transcriptomics

PMID: 34886989, PMID: 34268346, PMID: 33759339, PMID: 30592723, and PMID: 23641796.

PMID: 36230614, PMID: 34884478, PMID: 32620617, PMID: 29674676, PMID: 27177088, and PMID: 25964560.

(2) Please provide references in the legends of Figures 2 to 6. The same apply for Tables 1 and 2.

Author Response

Thank you very much for your efforts and comments. Below, we answer point to point.

As a result, the authors will need to revise as follows:

(1) Prior to thoroughly reviewing the present draft, I have to ask the authors to cross-check the below listed references for potential supplementation to the manuscript.

Epigenetics in canine lymphoma

PMID: 35955829, PMID: 35876604, PMID: 35409379, PMID: 34954594, PMID: 34570764, PMID: 32926463, PMID: 32154977, PMID: 28111882, PMID: 27747218, PMID: 27630997, PMID: 29061940, PMID: 25716962, and PMID: 25108839.

Oncogenes, tumor markers and transcriptomics

PMID: 34886989, PMID: 34268346, PMID: 33759339, PMID: 30592723, and PMID: 23641796.

PMID: 36230614, PMID: 34884478, PMID: 32620617, PMID: 29674676, PMID: 27177088, and PMID: 25964560.

Thank you very much for your effort and recommendations. Some of them has been included in the manuscript (see changes in red color). Other has not been included because the following reasons:

PMID: 35955829-This is an interesting study, but canine lymphoma is not included.

PMID: 35876604-It is not related to epigenetic alterations or drugs related it.

PMID:34570764- It is not related to epigenetic alterations in canine lymphoma.

PMID: 32154977-It is not related to specific epigenetic alterations in canine lymphoma, only with epigenetic signatures. It is an interesting study, but the review which we presented tries to explain specific changes in epigenetic regulation.

PMID: 28111882- Even though this study analyzes the differential expression through regulation by MEK, it does not specify the epigenetic mechanisms of this regulation, so it has not been included in the manuscript.

PMID: 29061940- This study does not speak specifically of epigenetic mechanisms. However, another publication from the same group (DOI: 10.2460/ajvr.75.9.835) in which these mechanisms are studied has been included in the review.

PMID: 25716962-This study analyses miRNAs related to canine mammary cancer, but not canine lymphoma.

PMID: 25108839-This study shows the results of chemotherapy, but it is not related to epigenetic mechanisms.

PMID: 34886989- Not related to epigenetic mechanisms.

PMID: 34268346- Not related to epigenetic mechanisms.

PMID: 33759339- Not related to epigenetic mechanisms.

These three studies are related to Ki-67 molecule, but not with its epigenetic regulation.

PMID: 30592723- Related to analysis of tumor progression, but not with epigenetic mechanisms.

PMID: 23641796- Related to VEGF and MMP as biomarkers in canine lymphoma, but not with epigenetic regulation.

PMID: 34884478- Related to inhibitors of cell proliferation treatment, but not with epigenetic mechanisms treatments.

PMID: 29674676- Related to transcriptomic dysregulation but not with epigenetic mechanisms.

PMID: 27177088-Related to indolylmaleimides treatment (inhibitors of cell proliferation) but not with epigenetic mechanisms.

PMID: 25964560- Related to biomarkers expression in vitro, but not with epigenetic alterations or mechanisms.

(2) Please provide references in the legends of Figures 2 to 6. The same apply for Tables 1 and 2.

Done.

Round 2

Reviewer 2 Report

Thank you very much for this very concise revision of the submitted manuscript - I am very fine with the revised version and I very much appreciate all the efforts made by the authors.

Only a few typing errors seem to be left for correction:

Line 117: tru-cut should be true-cut

Line 175: Please insert a semicolon after poor prognosis.

Line 392: Please insert space as follows: chemoresistant --> chemo resistant

Line 741: Please delete the dash in widely-studied.

Line 799: I guess c-Myb should be c-Myc.

1196: Please correct Blc-2 to Bcl-2

Author Response

Thank you very much for your corrections. We have implemented all suggested changes.

Line 117: tru-cut should be true-cut

Done.

Line 175: Please insert a semicolon after poor prognosis.

Done.

Line 392: Please insert space as follows: chemoresistant --> chemo resistant

Done.

Line 741: Please delete the dash in widely-studied.

Done.

Line 799: I guess c-Myb should be c-Myc.

Done.

1196: Please correct Blc-2 to Bcl-2

Done.